# Effect of Calcium Nitrate on the Properties of Portland–Limestone Cement-Based Concrete Cured at Low Temperature

**DOI:** 10.3390/ma14071611

**Published:** 2021-03-25

**Authors:** Gintautas Skripkiūnas, Asta Kičaitė, Harald Justnes, Ina Pundienė

**Affiliations:** 1Department of Building Materials and Fire Safety, Faculty of Civil Engineering, Vilnius Gediminas Technical University, Saulėtekio al. 11, LT-10223 Vilnius, Lithuania; gintautas.skripkiunas@vilniustech.lt; 2SINTEF Building and Infrastructure, Strindvegen 4, NO–7045 Trondheim, Norway; harald.justnes@sintef.no; 3Laboratory of Concrete Technology, Institute of Building Materials, Vilnius Gediminas Technical University, Linkmenų str.28, LT-08217 Vilnius, Lithuania; ina.pundiene@vilniustech.lt

**Keywords:** calcium nitrate, Portland–limestone cement, low temperature, setting time, compressive strength

## Abstract

The effect of calcium nitrate (CN) dosages from 0 to 3% (of cement mass) on the properties of fresh cement paste rheology and hardening processes and on the strength of hardened concrete with two types of limestone-blended composite cements (CEM II A-LL 42.5 R and 42.5 N) at different initial (two-day) curing temperatures (−10 °C to +20 °C) is presented. The rheology results showed that a CN dosage up to 1.5% works as a plasticizing admixture, while higher amounts demonstrate the effect of increasing viscosity. At higher CN content, the viscosity growth in normal early strength (N type) cement pastes is much slower than in high early strength (R type) cement pastes. For both cement-type pastes, shortening the initial and final setting times is more effective when using 3% at +5 °C and 0 °C. At these temperatures, the use of 3% CN reduces the initial setting time for high early strength paste by 7.4 and 5.4 times and for normal early strength cement paste by 3.5 and 3.4 times when compared to a CN-free cement paste. The most efficient use of CN is achieved at −5 °C for compressive strength enlargement; a 1% CN dosage ensures the compressive strength of samples at a −5 °C initial curing temperature, with high early strength cement exceeding 3.5 MPa but being less than the required 3.5 MPa in samples with normal early strength cement.

## 1. Introduction

Concrete is a composite material made up of cement matrix and aggregates. The gel, which is formed during the reaction between cement and water, solidifies and binds the aggregates. The most common binders for concrete are various types of Portland cements. According to the research, the cement industry relates to 5–8% of total human CO_2_ emissions [1,2] that is responsible for climate change [3]. One ton of cement produces about 900 kg of CO_2_ [4]. In this case of blended cements, energy losses and carbon footprint are reduced [5,6]. In global practice, the partial replacement of Portland cement with mineral additives in concrete has been used for a long time.

Limestone composite cement production requires less energy and emits less carbon dioxide, meaning that it has a better environmental performance than other types of cements [7]. The use of limestone to replace cement has been approved in many standards, such as the Canadian Standards Association (since 1983) [8], the European standard EN 197-1 (since 2000) [9], and ASTM C150 (since 2004) [10]. The usage of one generally accepted method allows us to adapt limestone as a partial replacement for clinker to obtain limestone–Portland cement in which, according to the European standard EN 197-1:2011, is allowed to replace 6–35% of clinker with limestone. In order to increase the use of limestone in cement production, cements with up to 65% of limestone [11] have been developed according to the limits of EN 197-1 and a concrete modification technology has been proposed [12]. Replacing 5–10% of cement with limestone provides only a similar or higher compressive strength than without [13]. The data presented reveal that concrete samples with up to 20% of limestone CEM II (A-L or A-LL) had similar values of compressive strength as CEM I 42.5 of concrete samples [14]. Studies reveal that, at a given water/cement ratio, concrete samples containing limestone could achieve a higher compressive strength and modulus of elasticity than the control samples made of pure Portland cement (CEM I) [15].

The influence of limestone on cement minerals is based on the delay in hydration of tricalcium aluminate during the first 16 h of hydration, which is the result of the chemical interaction between limestone and calcium aluminate hydrate and monosulfate [16], although some researchers hold opposite views [17]. Such a delay of the hydration process of Portland cement and possible decrease in the heat of hydration can have a negative effect on concreting at low temperatures. The use of limestone CEM II (A-LL) for concreting in low temperatures is problematic because not only do the setting times become longer but also the strength of the concrete decreases [18].

There are several known methods for concreting at low temperatures, such as heating the aggregate and water, the use of preventive measures, and insulation and heating of the concrete site. In contrast, the main danger in the production of concrete in winter conditions is negative temperature, which slows down the hydration processes, especially when ice starts to form. As it is pointed out in the research [19], the hydration rate of a cement sample at −5 °C temperature reaches only 16.7% compared to a sample cured at +20 °C temperature. As pointed out in [20] at temperatures lower than −5 °C, up to 92% of water, present in non-solidificated concrete, can transform into ice. As a result of such conditions, the strength evolution in such concrete was considerably retarded even considering that there is very little water that can react with cement. Although there is a gradual increase in strength as the concrete thaws, such concrete generally does not meet the requirements. In the event that the concrete mixture freezes, the strength properties of this concrete as well as its resistance to cyclic freezing–thawing decrease by 20–40% [21].

The use of high-strength cement or chemical admixtures can also be used to accelerate setting and to increase early strength [18,22]. A technologically simplified method is to lower the freezing point of the concrete mix. Anti-freezing admixtures for concreting can be used even at temperatures of −30 °C [23,24]. Set accelerating admixtures include urea, CN, calcium chloride, sodium nitrite, sodium chloride, potassium, and calcium chloride–nitrite–nitrate [22,25,26]. Admixtures such as calcium chloride, calcium nitrite, and CN, containing the same cations as alite and belite, accelerate hydration and thus activate hydrate crystallization processes [24,26]. CN in the paste [27] increases the concentration of calcium ions, leading to a faster calcium silicate hydrate (CSH) formation and reduces ettringite formation. Set accelerating admixtures increase the strength of concrete, lower the freezing temperature of cement paste, accelerate cement hydration processes [26,28], increase the surface area of hardened cement paste, and affect the pore structure of the hardened cement paste [23,28]. According to EN 934 [29], a set accelerating admixture should reduce the initial setting time by at least 30 min at +20 °C–27 °C and a maximum of 60% of the initial setting time of the control mix at +5 °C, determined at equal consistence of mortar. The concrete samples with a hardening accelerating admixture after 1 day at +20 °C–27 °C temperature should achieve a compressive strength of at least 120% when compared to the reference and a compressive strength of at least 130% when compared to the reference value after 2 days at +5 °C at equal consistence of mortar (EN 934 [29]). CN and NaNO_3_ are often used together with CaCl_2_ as setting accelerators and has an impact on the hydration of Portland cement [27,30,31,32]. The results from several investigations demonstrated that the chloride salts accelerate hardening and setting times; other nitrite salts work as setting accelerators [24]. NaNO_3_, K_2_CO_3_, and Na_2_SO_4_ promote the hydration of cement; however, they possess different chemical mechanisms of acceleration [22,32].

Today, most suppliers of admixtures offer accelerators based on CN [33,34]. CN like non-chlorine set accelerating salts was patented in 1969 [35]. CN makes an impact on concrete properties such as multifunctioning admixtures: set accelerator, anti-freeze admixture, inhibitor, long-term strength enhancer, and counteraction of retardation by plasticizers while maintaining rheology [34,36,37,38]. Justnes et al. investigated at +5 °C, +13 °C, and +23 °C the final and initial setting of cement pastes with 1.55% CN. They found that it works better at lower temperatures [36]. Ramachandran found that CN at low temperatures and concentrations acted as a setting time accelerator in cement [32]. According to El-Didamony and other researchers, the set acceleration of CN increases with increasing belite content in cement [39].

Dong et al. investigated the workability and strength increase in concrete when curing at several temperatures (−5 °C, −10 °C, −15 °C, and −20 °C) and under standard conditions using different admixtures such as water-reducing accelerators, chlorine-free antifreeze, and air-entraining admixtures [40]. They argued that admixtures can shorten setting times, can ensure the sufficient strength of concrete early, can inhibit negative temperature effects of concrete, and can facilitate thermal storage of concrete. The researchers studied the use of urea and CN in concreting in cold conditions [23,24,25]. Concrete samples with 6% CN admixture and without at −5 °C, −10 °C, −15 °C, and −20 °C were investigated. The CN significantly affected the early age compressive strength [24,40]. The compressive strength increases with the amount of admixture. It is stated that CN can be used for cold concreting without any additional measures. Antifreeze admixture CN influenced the hydration process according to [41]. The best physical and mechanical properties of the concrete mixtures were obtained when 3% calcium nitrate and 5% hydroxyethylamine mixtures were used.

The use of limestone CEM II (A-LL) cements for concreting in cold weather is virtually unexplored and their interaction with CN is insufficiently studied. That is why, for this study, two limestone cements, CEM II A-LL 42.5R and CEM II A-LL 42.5N, are studied. This article analyses the influence of CN on the properties of fresh cement paste hardening processes and hardened concrete properties at different initial curing temperatures.

## 2. Materials and Methods

High early strength and ordinary early strength limestone–Portland cements from UAB AKMENES CEMENTAS (Akmene, Lithuania) were used for the research. Both cements were produced during the same period, and the same clinker was used in the production of both cements.

The main properties and mineralogical composition of cements are presented in Table 1 and Table 2.

Fine aggregate (sand of 0/4 fraction) and coarse aggregate (gravel of 4/16 fraction) were used for concrete production. Both aggregates met the requirements for concrete of the standard EN 12620:2003. The used water complied with the requirements of the standard EN 1008:2005.

All material used in the studies and the stirring of pastes were carried out at a temperature of +20 °C. Water for a normal consistency cement paste has been determined at the temperature of +20 °C too. The w/c ratio in cement pastes was 0.24 for CEM IIN and 0.267 for CEM IIR pastes. The compositions of fresh cement pastes for investigating viscosity and setting times are presented in Table 3.

Cement pastes with CN dosages from 0 to 3% (from cement mass) were prepared for the tests. The CN was dissolved in water before mixing. The effect of CN on the cement paste setting times (initial and final) were tested using the Vicat device (Controls, Milan, Italy). The setting was registered by penetrating the cement paste with a needle using constant force. The sample was tested every 10 min. The initial setting time was recorded when the needle does not penetrate the entire thickness of the sample, and the final setting time was recorded when the needle can no longer penetrate the sample. The Vicat device was stored in a temperature-controlled freezing chamber at +20 °C, +5 °C, 0 °C, −5 °C, and −10 °C temperatures. For the measurements, the Vicat device with paste sample was shortly (about 10 s) removed from the chamber and returned back to the chamber immediately after the measurements. The measurements took place at ambient temperature (20 ± 2 °C).

The dynamic viscosity of cement pastes with different amounts of CN was studied with an SV-10 vibro-viscometer (from A&D, Tokyo, Japan (SV-10)). The measuring limit of this device was 12,000 mPa·s; the measuring accuracy reached 0.01 mPa·s. Cuvettes of 45 mL volume were used for the tests. The volume of cement paste sample was 35 mL. The device measures the resistance of the paste viscosity to constant vibration of gauge plates at 30 Hz frequency. The proportional to viscosity resistance force was transformed into an electrical signal and registered. The dynamic viscosity of the prepared pastes was immediately measured (5 min after the addition of water to the cement) and then after 10, 20, and 30 min. The temperature of the pastes was +20 ± 3 °C.

Select concrete mixes were prepared in the laboratory. Before concrete mixing, the CN was dissolved in water. First, the dry mixture of cement, and fine and coarse aggregates were mixed in a pan-type laboratory mixer for 1 min. Water with CN or without it was then added and stirred for 3 min. The amount of CN varied from 0.5 to 3% from the cement mass. The amount of water and cement in all mixes was the same. The v/c for all mixtures was the same: 0.55. In all mixtures, an amount of 0.5% from the cement mass of the superplasticizer was used. The quantity of raw materials is presented in Table 4.

The slump and the density of the concrete mixture were evaluated according to EN 12350-2 and EN 12350-6. The entrained air content was measured from the concrete mixture by EN 12350-7 using the pressure gauge method and equipment. Concrete samples were placed in molds and cured for 2 days at selected temperatures (−10 °C to +20 °C) in the freezing chamber, and then for another 26 days, they were cured in water at +20 °C. Samples (100 mm × 100 mm × 100 mm) molded from the prepared mixture were kept in molds for 2 days in different conditions in the chamber RUMED 3001 (Controls, Milan, Italy) (at select temperatures +20 °C, +5 °C, 0 °C, −5 °C, and −10 °C) and, then for 26 days, cured in water at a temperature of +20 °C. The control sample groups after demolding were held in water curing for 28 days at ambient temperature (+20 ± 2 °C).

Hardening of the samples was carried out according to EN 12390-2; density and compressive strength were measured according to EN 12390-7 and EN 12390–3, respectively. Each of the 6 samples from each batch was compressed, and their total average was calculated.

## 3. Results

### 3.1. Cement Paste Viscosity Test

The use of a vibro-viscosimeter to measure the viscosity of cement pastes is very useful because it allows us to observe the changes in cement paste viscosity at any interval until the viscosity of the paste reaches the limit value of the instrument readings. Changes in dynamic viscosity represent degradation of rheological properties, loss of workability during the hydration process, and the development of new phases. The results of the CN dosage impact on CEM IIR (high early strength) and CEM IIN (ordinary early strength) cement pastes’ viscosities are presented in Figure 1 and Figure 2 during times up to 30 min.

Immediately after paste preparation, the viscosity of control CEM IIR and CEM IIN pastes differed significantly. CEM IIR paste viscosity is 14.7% lower than the CEM IIN paste viscosity (Figure 1 and Figure 2). This difference is partly due to the higher w/c in the CEM IIR paste. Another reason is that CEM IIN contains significantly more minerals C_3_A and C_3_S, which are immediately involved in the hydration process and increase the viscosity of the paste [42]. A similar difference persists after 30 min of measurements.

Increasing the CN amount in CEM IIR from 0.5% to 1.5% immediately after mixing decreases the viscosity of the paste from 11.3% to 24.6% relative to the reference (LR in Figure 1). A paste containing 2% CN increases the viscosity of the paste up to 24.2% when compared to a paste with 1.5% of a CN admixture. Larger amounts of CN (2.5–3%) increase the viscosity of a paste by 7.05–12.2% in comparison to the control sample and by 41.9% and 48.8% in comparison to a paste with 1.5% CN. The same tendency remains when increasing the time after mixing. In pastes with larger amounts of CN (2–3%) after 30 min, the viscosity of pastes is 18.5–38.3% higher than that for the control paste. The viscosity of pastes with less (0.5–1.5%) CN is higher than in that in the control paste by 4.0–8.0% after 30 min from mixing. We can conclude that higher amounts 2–3% CN promote the significant growth of viscosity of pastes. It can be supposed that a higher mixing water content is required to achieve normal consistency of the CEM IIR paste, which can induce a more active reaction of CN with C_3_A and C_3_S.

An increase in CN in the CEM IIN paste from 0.5%, 1.0%, 1.5%, and 2.0% decreases the viscosity of paste by 11.1%; 13.3%, 15.3%, and 10.2%, respectively, immediately after mixing. When the amount of CN is increased to 2.5% and 3%, viscosity starts to increase compared to the control sample LN in Figure 2 until 2.4% and 9.4%. Greater differences in comparison to CEM IIR pastes are observed after 20–30 min, when the viscosity in pastes with more (2–3%) CN increases until 4.9–22.6% in comparison to the control CEM IIN–type paste sample. The viscosity of pastes with less (0.5–1.5%) CN is higher by 0.63–3.2% than that in control pastes.

We can conclude that lower amounts up to 2% CN reduces the viscosity of pastes but that amounts more than 2% demonstrate the effect of increasing viscosity. This may be due to the increase in paste’s temperature: with an increase in CN amount in the paste, the temperature of paste increases significantly via high CN dosage. This effect has been described by Kicaite et al. [43], where the exothermic profile of cement pastes with CN amounts 0–3% was tested. It was obtained that CN noticeably increases paste temperature during the first 30 min (up to 31 °C in pastes with 3% CN), reduces the induction period, and fastens the exothermic reaction (EXO) maximum time. Additionally, higher paste temperatures also promote faster hydration of cement minerals and result higher mechanical properties [19,44,45].

The second reason can be granulometry of both cements. For CEM IIR pastes, the smaller cement particles can induce a more active reaction between CN and the cement minerals C_3_A and C_3_S. We also can see that, in the presence of higher amounts of CN, the growth of viscosity in CEM IIN pastes is much slower than in the CEM IIR pastes.

### 3.2. Setting Time of Cement Paste with CN at Different Temperatures

The setting time tests were performed by varying the amount of CN in the paste from 0 to 3% and the test temperature from +20 °C to 0 °C (Figure 3, Figure 4, Figure 5 and Figure 6).

At +20 °C, CN shortens the initial and final setting times of both cement pastes (Figure 3, Figure 4, Figure 5 and Figure 6). Shortening the final setting time is more effective when using large amounts of CN. The difference between the initial and final setting times is more than 50 min for control cement pastes (58 min in the case of CEM IIR and 54 min in the case of CEM IIN) and about 35 min for both type pastes with 3% CN (37 min for CEM IIR and 30 min for CEM IIN). The relationships, obtained during hydration course, presented in Kicaite et al. [43] are confirmed. However, the results of the study [25] show that, at temperatures between +7 and +20 °C, CN works as a set accelerator for cement paste.

The initial and final setting times of cement pastes increase when the temperature of cement pastes drops to +5 °C. The initial setting time of the control CEM IIR paste and CEM IIN paste increases by 141 min and 75 min accordingly. It occurs because decreasing rates of the chemical reaction in lower temperatures are observed, and the initial setting is delayed [19]. As referred to in [43], most of the chemical reactions, as a rule, accelerated twice, when the temperature increased by 10 °C. Adding 1% CN into the paste results in shortening of initial setting time by 64.1% in the case of CEM IIR paste and by 81.3% in the case of CEM IIN paste when compared to the setting times of both cement control pastes without CN. In this way, CN dosages of 1.3% (Figure 3 and Figure 4) can be used as a set accelerator for CEM IIR because, according to EN 934-2, a set accelerator is an admixture that achieves 60% of the initial setting time values of control paste. For the CEM paste with 1% CN, the paste reaches 64% of the initial setting time of control paste. The presented results revealed that the set accelerating efficiency of CN depended very much on the cement type, as is in [30]. The research concluded that, at temperatures of +5–7 °C, the efficiency of CN as a set accelerator increases with increasing C_2_S in cement. However, our study shows opposite results because the C_2_S amount in CEM IIR is 1.5 times greater than in CEM IIN.

At +5 °C, the final setting times of the control CEM IIR paste and CEM IIN paste increase by 192 min and 164 min accordingly, in comparison to +20 °C; 1% CN in the paste results in shortening of the final setting times by 82.2% in the case of CEM IIR paste and 98.4% in the case of CEM IIN type paste when compared to the setting times of control cement paste. The shortest final setting times are obtained by 3% CN dosage. This amount shortens the final setting of CEM IIR paste and CEM IIN paste by 326 min and 267 min, which are 21.4% and 28.4% of control cement paste without CN. The study carried out by [6,36] shows that the efficiency of CN is more pronounced at lower (+5 °C) temperature.

At 0 °C, the setting times of pastes are further extended. The initial setting times of control CEM IIR paste and CEM IIN paste increase by 175 min and 77 min compared to 20 °C values. The final setting times for both pastes increase to 384 min and 322 min. When 3% CN was added, the final setting times of the CEM IIR and CEM IIN pastes shortened, respectively, to 21.4% and 28.4% of control cement paste. Other studies on the investigation of setting times corroborate these findings [46].

At ambient temperatures below 0 °C, water-freezing processes predominate and cement hydration processes slow down considerably, making it difficult to accurately distinguish between the binding processes using standard test procedures. In this case, it is difficult to define the basic “setting”; this process can be called a paste solidification process. The initial and final solidification times for CEM IIR and CEM IIN cement pastes at −5 °C and −10 °C are given in Table 5 and Table 6.

With the increase in CN, a decrease in the initial solidification time is observed at −5 °C for CEM IIR paste. Meanwhile, we do not observe such an effect for CEM IIN paste. In the case of the final solidification time, we also observe that the use of CN for CEM IIR paste shortens this time. Studies at −10 °C show the same tendencies that the initial solidification time shortens with an increase in CN amount. Compared to the test results observed at −5 °C, shortening of the solidification time is more pronounced. However, as noted, at temperatures below 0 °C, it is difficult to distinguish freezing and setting processes. The gradual freezing of the mixing water can be more responsible for the needle penetration reduction than the cement setting processes and obtained freezing-setting phenomenon. Based on this assumption, it can be concluded that the method used does not allow for proper testing of cement paste at a temperature below 0 °C.

### 3.3. Technological Properties of Concrete Mixture with CN

Concrete mixtures with CEM IIR- and CEM IIN-type cement were tested immediately after mixing and after 1 h (Figure 7). CEM IIR- and CEM IIN-type cement paste slump test results performed immediately after mixing are marked in Figure 7 as 0 h, and paste slump test results, performed after 1 h, are marked as 1 h. The slump of both concrete mixes increases when the CN amount is raised from 0% to 1% (Figure 7). The introduction of 0.5% CN and 1% CN in the concrete mix with CEM IIR increases the slump by 10–30 mm and by 40–60 mm accordingly, but in Figure 7, the slump values are presented as the average of four measurements. The introduction of 0.5% CN and 1% CN in the concrete mix with CEM IIR increases the slump by an average of 20 mm and by an average of 45 mm, respectively. The same trends were observed in the research of [19]; 1.7% CN increases the concrete mix slump up to 30% compared to the reference concrete mix slump. Meanwhile, 3% CN shows a 10 mm decrease in slump compared to the slump of the control concrete mix. The above trends are observed immediately after mixing and correlate well with the CEM IIR paste viscosity tests performed (Figure 1).

Similar trends are observed in the case of concrete mix with CEM IIN cement. The introduction of 0.5% CN and 1% CN in the concrete with CEM IIN increases the slump of the concrete mix by 23–48 mm and by 33–53 mm accordingly; however, the same slump values are presented as the average of four measurements in Figure 7. This effect is more pronounced when compared to the concrete mix containing CEM IIR. Additionally, the introduction of 3% CN in the concrete increases the slump to 8.84% compared to the slump of a control concrete mix. The above trends are observed immediately after mixing and correlate with the viscosity tests presented in Figure 2. We can conclude that up to 2% CN works as a plasticizer and that higher than 2% CN in the concrete mix reduces the slump. CEM IIN paste with a CN amount of 3% has been proven to have a lower viscosity than an analogous CEM IIR paste with the same CN amount.

After one hour, the slump of the both concrete mixes decreased almost 3 times, but the previously observed trends remaining for −0.5% and 1% CN in the concrete mix demonstrate the highest slump values. In general, it can be observed that mixes with CN slumps are higher than CN-free mixes [23,24].

The effect of CN on the entrained air content in concrete is plotted in Figure 8. The tested concrete was mixed with a polycarboxylate ether-based super plasticizing admixture with reduced content of anti-foaming agents. That type of admixture provides required quantity of entrained air content (4–6%) for freezing–thawing resistance concrete without air–entraining admixture. The air content in the concrete produced with CEM IIR decreases from 5.8% to 4.8% when 1% CN is added. Further increasing the CN amount to 3%, the air content remained virtually unchanged at 5.1%. Different effects of CN amount are observed when concrete contains CEM IIN cement. It can be seen in Figure 8 that the control concrete mix entrains 6.6% air and that this is 13.8% more than in the control concrete with CEM IIR cement mix; 0.5% CN slightly increases the amount of entrained air to 7.1%. With further increases in CN, the content air entrained decreases proportionately to 4.4%.

The density of the concrete mixture and the air content are related. The density of the control concrete mixture was 2336 kg/m^3^, and the maximum value was reached at a CN amount of 1% 2375 kg/m^3^ for CEM IIR. Higher (2 and 3%) CN amounts did not affect the density of the concrete mixture. The effect of CN amount on the density of a concrete mixture with CEM IIN is more pronounced. The minimum value for a CN amount of 0.5% was 2318 kg/m^3^ and the maximum value for a CN amount of 2% was 2366 kg/m^3^, which correlates with the results of the air content testing.

### 3.4. Compressive Strength of Concrete with CN in Ordinary at Low Temperature

For hardening at initial temperatures +20 °C, the best compressive strength results for samples with CEM IIR are obtained using 1% CN (Figure 9). In comparison to control samples after 2 days of hardening, the compressive strength of concrete increases by 17.5%; after 7 days, by 21.5%; and after 28 days, by 19.6%.

Other trends are observed with the addition of CN with ordinary early strength cement CEM IIN. It can be observed that CN has a smaller effect on the early strength (2 and 7 days) of concrete with CEM IIN (Figure 10). The results reveal that 1% CN alone acts as an accelerating admixture; however, it has only a minor positive impact on the long-term growth of mechanical strength, which is confirmed by Polat [18,47]. The most effective amount in concrete in terms of strength is 3% CN. The compressive strength of concrete increased by 14.1% after 2 days of hardening, by 12.4% after 7 days, and by 32.47% after 28 days compared to control samples. It can be concluded that, at +20 °C for concrete with CEM IIR and for concrete with CEM IIN, 1% and 3% CN, respectively, are most effective. The CN amount (0–4%) efficiency for different cement types CEM I and CEM II/A-LL was tested in research [48]. The samples were cured for 7 days under water and further until testing at +20 °C/65% relative humidity. The compressive strength results after 28 days of hardening show that, for CEM I samples, increasing amounts of CN increased by 12.7% until 4% compressive strength of the samples and that, for CEM II/A-LL samples, compressive strength increased by 22.2%. The same verification of the compressive strength enhancement was identified in [49]. Additionally, it was pointed out that the compressive strength enhancement is related to modification in the porosity of samples.

The influence of the initial curing temperature +5 °C on the compressive strength of concrete samples with CEM IIR is presented in Figure 11. It can be pointed that, after 2 days of initial hardening at +5 °C, the compressive strength of the reference concrete sample, cured at temperature +5 °C, is more than 2 times lower than that sample prepared and cured at +20 °C temperature. Such a difference can be explained by the different degrees of hydration in cement paste cured at different temperatures. As illustrated by the research [19], the degree of hydration in cement paste samples significantly depends on the sample curing temperature: for example, the degree of hydration for the same samples cured at +20 °C temperature for 1 and 3 days increased from 48% to 68%; for same time, cured samples at +8 °C increased from 38% to 62% and the samples cured at +5 °C increased from 35% to 58%.

After 7 days of hardening, the best compressive strength results were established for concrete samples with 2% CN and the strength values are 18.4% higher compared to the reference sample. After 28 days of hardening, the best compressive strength results were obtained, with 3% CN being 28.7% higher compared to the reference.

The same tendencies were observed in [45,50], where it is concluded that cement pastes without CN, cured at lower +4 °C temperature, have a lower strength than those cured at higher +20 ± 2 °C.

The initial hardening temperature has a very high effect on the early strength of concrete. It is important to note that samples that were kept at low temperatures for 2 days and then cured at +20 °C in water show higher compressive strength results after 28 days of curing than reference concrete samples cured at +20 °C in water at all the time. For the samples with CN contents of 1, 2, and 3%, the compressive strength results were 6.9, 1.9, and 12.8% higher, respectively, compared to the samples that were not kept at low temperatures (reference samples).

The compressive strength of concrete with CEM IIN is more strongly affected by 2% and 3% amounts of CN (Figure 12). After 2 days of initial hardening at +5 °C, the compressive strength of samples with 3% CN is 11.4% higher compared to reference samples without CN. After 7 days of hardening, the compressive strength with 2% CN was 9.1% higher and that after 28 days was 4.7% higher than the reference. Studies have shown that, in all cases, two-day storage gives higher results compared to samples that were not initially stored at low temperatures.

The data in Figure 10 and Figure 12 show that the effect of CN is more pronounced for concrete samples with CEM IIR initially hardened at +5 °C compared to reference hardened at +20 °C. For concrete with CEM IIR and CEM IIN, the most effective CN dosage seems to be 3% and 2%, respectively. This may be due to the higher belite content of the CEM IIR (Table 1), which according to El-Didamony et al. [30,39] has accelerated hydration processes in the presence of CN.

The compressive strength of specimens with CEM IIR after 2 days of initial curing at 0 °C is equal to or exceeds 3.5 MPa, as shown in Figure 13, and meets ACI 306R–10 requirements [41]. After 2 days of initial hardening, the increase in compressive strength of the samples with 1% CN reaches 302.9% compared to the compressive strength of the reference samples without CN. After 7 and 28 days of hardening, the increase in compressive strength reaches 30.9% and 16.9%, respectively, in comparison with reference samples. However, it should be noted that the best results, reached for samples with 1% CN after 2 days of initial hardening, is up to 2.5 times lower than the compressive strength of samples cured at +20 °C for 2 days (Figure 9). The research [50] examined cement samples, cured for 2 days at (0 °C, −5 °C, −10 °C, −15 °C, and −20 °C) temperature and then 26 days cured in water at +20 ± 2 °C temperature. In samples containing 1% CN, the compressive strength varies from 23.24 MPa to 14.8 MPa with increasing negative curing temperature. It may be noted that the degree of hydration in cement paste samples cured at 0 °C temperature [19] is not high and reaches 28% after 1 day of curing and 43% after 3 days of curing.

Meanwhile, after 7 days of hardening, the increase in compressive strength reaches only 9.3% in comparison to samples cured at 20 °C. Almost in all cases, higher values are found after 28 days when the samples are stored at low temperature for 2 days and reach 11.5% compared to the samples that have been cured at +20 °C all the time. The best results are when 0.5 and 1% CN are used.

When CEM IIN was used in concrete composition, higher compressive strength results are achieved for concrete specimens when 2% CN is used (Figure 14). After 2 days of initial curing 0 °C temperature, the compressive strength is 27.5% higher than compared to the reference without CN and reaches 5.2 MPa. In this case, a lower efficiency of CN is observed compared to concrete samples with CEM IIR. After 7 days, the highest compressive strength values are reached with 3% CN and are equal to the compressive strength of the reference samples. It is important to note that the compressive strength of samples with 2% CN initially cured at 0 °C for 2 days are 2.8 times lower than for samples with the same composition cured at +20 °C for 2 days (Figure 10). However, after 7 days of curing, the compressive strength values in these samples are 9.6% higher than in the same composition samples cured at +20 °C for 7 days (Figure 10).

Figure 14 demonstrates that, after 28 days, higher compression strength values are achieved with 0.5–2% CN, but in general, the results obtained are approximately 10% higher in a comparison with samples hardened 28 days at +20 °C.

It can be concluded that with the addition of CN, additional hardening in water is necessary to achieve a high compressive strength result. Karagöl [24] came to the same conclusion after observing concrete with various freezing times and subsequent immersion and curing in water for up to 28 days. When CN in concrete was used, the author pointed out that additional water curing was needed for the production of compressive strength because water curing would vitalize the frozen cement paste, allowing it to regain its original compressive strength.

At −5 °C temperature, the reference CN-free samples with CEM IIR and samples with 0.5% CN did not reach the necessary level of strength (Figure 15). The value of compressive strength after 2 days of initial hardening at −5 °C exceeds 3.5 MPa and reaches 6.1 MPa even when 1% CN was used. In this case, no additional protective measures of concrete are needed to achieve the required strength; all that is required is a sufficiently humid atmosphere for further hardening. After 7 days, the highest compressive strength values are reached with 3% CN, giving 262.8% higher than the compressive strength of the reference samples without CN. According to the authors [19], at negative temperatures, such as −5 °C, the hydration process of cement minerals continued slowly because mineral ions dissolved in water, such as Ca^2+^, K^+^, Na^+^, OH^−^, and SO_4_^2−^, prevented ice formation to some extent and lowered the freezing point of water [51,52]. The authors proved that, in the cement paste samples, cured at −5 °C temperature, the degree of hydration reaches 16.7% after 1 day and 25.5% after 3 days, although it is significantly lower in comparison with samples cured for the same duration at +20 °C: 48 and 68%. This is why the compressive strength of cement paste samples cured at negative temperatures is low. However, at the same time, it indicates that the cement can still hydrate at −5 °C. However, CN, used in dosage of 6%, at low temperatures has a similar accelerating effect as pointed out in [24], mostly due to the early formation of portlandite in cement stone [53]. Higher amounts of CN, 9 wt.% in the cement paste and a combination of CN 4.5 wt.% with urea 4.5 wt.%, were tested at low temperatures ranging from −5 °C to −20 °C [54]. The samples with a combined admixture of CN and urea show the highest compressive strength values after curing at −5 °C temperature after 7 and 28 days: 38.79 MPa and 41.91 MPa. After the same period, for samples with 9% CN, the compressive strength values are 19.4% and 1.8% lower. The control samples without additives present just 7.92 MPa and 3.12 MPa at this period. The same composition samples cured at −20 °C show significantly lower compressive strength values, but the compressive strength of samples with 9% CN is the highest, 16.01 MPa and 4.63 MPa; that for samples with a combined admixture of CN and urea is 12.79 MPa and 3.99 MPa; and that for control samples us 6.55 MPa. In contrast, the study conducted by [18] with 2 parties of concrete mixes containing 6% CN or 6% urea at −10 °C estimated that, after 28 of curing, the samples containing 6% CN reached a compressive strength value of 28.05 MPa, while the samples containing 6% urea reached only 18.32 MPa. The main reason why the CN admixture improves the development of strength is that CN contains Ca^2+^ ions, as in C_3_S and C_2_S minerals; this is why CN accelerates the hydration process in the early stage with faster hydrates forming, and in the later stage, CN can lower the eutectic point, which can play a significant role in the evolution of strength.

When comparing the compressive strength results for samples with 0.5% and 1% CN cured at −5 °C for 2 days with compressive strength of samples cured at +20 °C (Figure 9), it can be stated that the compressive strength of samples cured at −5 °C is 7.7–8 times lower than in the equivalent samples cured at +20 °C. However, after 7 days of curing, the compressive strength values in these samples are 55−10% lower than in samples of the same composition cured at +20 °C (Figure 9).

After 28 days, a higher compressive strength can be observed with 0.5 and 1% CN, but these results are significantly lower than those for samples hardened for 28 days at +20 °C and especially when compared to compositions without CN, where the difference is almost 2 times.

Concrete specimens with CEM IIN do not reach the required values of 3.5 MPa after 2 days of initial hardening at −5 °C (Figure 16). It is necessary to note that the compressive strength of the same composition samples cured at +20 °C reached 18–21 MPa after 2 days (Figure 10). This result may be influenced by a lower amount of C_2_S in CEM IIN than in the CEM IIR and suggests that slower hydration responds to slower strength growth.

However, after 7 days of hardening, 1% CN improves the compressive strength values more than 2 times (up to 35 MPa) compared to the compressive strength values of the CN-free reference (17 MPa). In comparison to the compressive strength of concrete with the same composition cured at +20 °C for 7 days, it can be observed that the compressive strength is the same.

After 28 days of hardening, the most effective dosage is 2% CN, giving a compressive strength at 88.9% higher than the compressive strength values of the reference CN-free sample, and it can be observed that these results are equal to the results in samples hardened for 28 days at +20 °C. After 28 days, higher compressive strength values can be observed with 0.5, 1, and 3% CN, but these results are noticeably lower than samples hardened for 28 days at +20 °C. Especially when compared to compositions with 0.5% and 1% CN, this difference is almost 2 times.

Compressive strength results of concrete samples, initially cured at −10 °C temperature, are presented in Figure 17 and Figure 18. The samples with both cements and 0.5–3% CN after hardening for 2 days at –10 °C did not reach the required value of 3.5 MPa. In addition, it is known that the eutectic point of CN ranges between −7.6 °C and −11.5 °C [55] and that curing at −10 °C temperature is very close to the specified lowest eutectic point of CN. It can be assumed that, where the ambient temperature drops by 10 °C, the reaction rate slows down by 2 times [56]; based on the calculations made by [19], it can be predicted that the degree of hydration will be 10–13% after 1 day of curing at −10 °C temperature. This indicates that the use of CN alone at this temperature is not sufficient and that additional implements are needed. Another possible method is the use of a larger amount of CN in the composition. Concrete mixtures with 6% CN [20] ensure 33.21 MPa, 10.76 MPa, 5.35 MPa, and 4.13 MPa compressive strength values for the samples cured for 28 days at −5 °C, −10 °C, −15 °C, and −20 °C. The compressive strength improves by, respectively, 1.73, 4.92, 9.59, and 11.38 after additional curing of the samples in water for 28 days. In a similar study [53], it was concluded that 6% CN resulted in a compressive strength of concrete samples in the range of 31.45 MPa–15.53 MPa at −5 and −20 °C temperatures compared to the compressive strength equal to 7.92–6.57 MPa with CN-free samples. It was concluded that the addition of CN at 6% increases the compressive strength of concrete samples by 297% when cured at −5 °C and by 96% when cured at −20 °C.

However, further storage of these samples in water ensures hydration of the cement, which significantly increases the compressive strength of the concrete after 7 and 28 days of hardening in water. After 7 and 28 days of hardening, the best results for concrete samples with CEM IIR are obtained with CN at 3%. For the concrete samples with CEM IIN, the best results after 7 days of hardening are obtained with CN at 3%, and after 28 days of hardening, the best results are obtained with CN at 1%. The samples, cured for 7 and 28 days at +20 °C, show approximately 2 times higher compressive strength than concrete samples initially cured at −10 °C.

In summary, the effect of CN decreased with decreasing initial curing temperature, but additional water curing (especially for 28 days) significantly increased the compressive strength of all the samples.

## 4. Discussion

In order to better evaluate the influence of dosages of CN on concrete compressive strength development at different temperatures, the obtained compressive strength results after 2 and 28 days of hardening are summarized in Figure 19, Figure 20, Figure 21 and Figure 22.

It can be seen that, after 2 days of hardening (Figure 19 and Figure 21), the optimal dosage for concrete with CEM IIR is 1% CN. This amount ensures higher compressive strength when with samples are initially cured down to −5 °C. For the concrete samples with CEM IIN, the optimal dosage of 3% CN ensures higher compressive strength down to 0 °C temperature. At lower temperatures of −5 °C and −10 °C, all tested CN amounts did not provide sufficient compressive strength of the samples.

The CN admixture forces the water within the sample to remain at least partially liquid until the temperature falls below the “eutectic point”. It can be supposed that, if curing of samples occurs in temperatures that are near or below the eutectic point of CN degree of hydration drops, the compressive strength of the samples significantly falls due to the formation of ice in pores and it results in microcracks, as is reported in [55]. As it is pointed in the research [19], at low temperatures (−5 °C), particles in the cement paste did not interact or bind with others because particles in the cement paste were moved and separated by ice formed at −5 °C and, as a result, the spaces between particles were not filled by hydrates.

When CN is used in the composition, a denser microstructure of the samples is achieved, which is confirmed by our sample density studies. When samples with CN are cured at lower temperatures (−5 °C and −10 °C), more C–S–H gel and portlandite are formed in the structure [23]. When the duration of the curing time extends, the formed ettringite gradually decreases and more C–S–H gel is formed, leading to a dense microstructure [57]. More in-depth insights into the hydration processes of cement and CN are presented in [26]. The CN solution, in contact with C_3_S and C_2_S, changes the ionic strength and pH of the solution and increases the density of the pore solution, resulting in the formation of hydroxysalts. Hydroxysalts promote densification and a change in microstructure of cement stone.

For concrete samples based on CEM IIR (Figure 20) hardened for 28 days in the temperature range 0 °C–(+20 °C), the most effective CN dosage is also 1%. However, at lower the temperature (−5 °C and −10 °C), the higher amount of CN is necessary to achieve higher strength values. The efficiency of CN depends directly on its content: the higher the dosage, the higher the strength values achieved.

For concrete samples with CEM IIN (Figure 22), the optimal 2% dosage of CN ensures higher compressive strength values down to −5 °C for samples that were hardened for 28 days. At the lowest temperature −10 °C, the efficiency of CN depends directly on its amount. Additional water curing (especially for 28 days) significantly increases the compressive strength of all the samples because melting of the ice found in the samples promotes the hydration of cement minerals.

It can be concluded that in temperature interval 0–(+20 °C), for CEM IIR significantly lower amount of CN is necessary than for CEM IIN. This is related with significantly higher increase of viscosity in CEM IIR pastes and faster hydration processes for belite reaction induced by CN. At −5 °C and −10 °C temperature, the efficiency of CN depends directly on its dosage and these trends are common in both cements.

## 5. Conclusions

Larger amounts of CN (2 and 3%) increase the viscosity of both CEM IIR and CEM IIN cement pastes. The effect is more pronounced for CEM IIR cement paste. A CEM IIN paste with 3% CN exhibiting lower viscosity than an analogous CEM IIR paste with the same CN amount. The CEM IIR cement granulometry and higher w/c required to achieve normal consistency caused by smaller cement particles of CEM IIR compared to CEM IIN can explain the more rapid increase in viscosity.CN contents at 0.5% and 1% in the cement pastes increases the slump of concrete. This tendency does not change during the first hour. The slump of concrete mix, regardless of the type of cement, decreases when the CN amount increase above 1.5%. These slump results for concrete correlates well with viscosity studies on pastes.The accelerator efficiency of CN increases with deceasing temperature from +20 °C to 0 °C. CN is the most effective as an accelerator at +5 °C and 0 °C. At these temperatures, the use of 3% CN reduces the initial setting times for CEM IIR paste by 7.4 and 5.4 times, respectively, and for CEM IIN paste by 3.5 and 3.4 times when compared to a CN-free control paste.Reductions in the compressive strength of concrete with CN were lower than that of the control without CN when initially cured at temperatures lower than +20 °C. The early strength of samples with CEM IIR cured for 2 days at +20 °C, +5 °C, and 0 °C when 1% dosage of CN was used in the concrete, while 3% dosage of CN was required for CEM IIN. The most efficient use of CN is achieved at −5 °C, where 1% CN ensures the compressive strength of samples with a CEM IIR higher than 3.5 MPa but less than the required 3.5 MPa in the samples with CEM IIN. The samples with both cement and 0.5–3% CN after hardening for 2 days at −10 °C did not reach the required value of 3.5 MPa.According to the procedure where concrete samples are hardened for 2 days in cold conditions and further hardened at +20 °C for 26 days, the compressive strength of the concrete samples is higher than that of the samples hardened in water at +20 °C continuously for 28 days.

## Figures and Tables

**Figure 1 materials-14-01611-f001:**
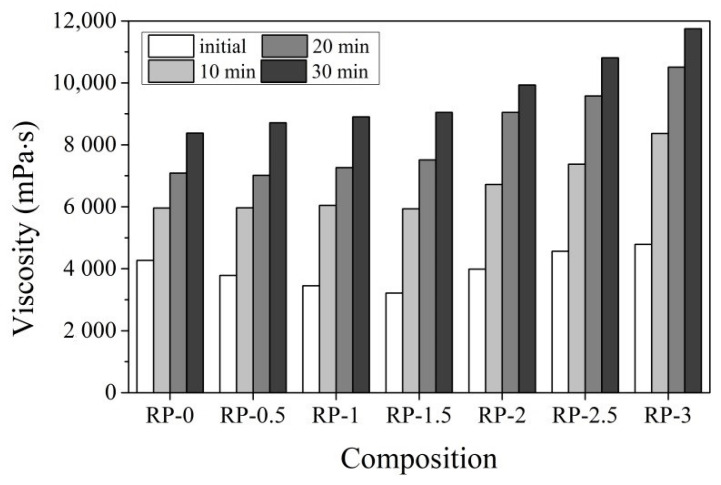
Viscosity of CEM IIR paste (LR) with different dosages of CN (0–3%) depending on time at +20 °C temperature (0—immediately after paste preparation; 10—after 10 min; 20—after 20 min; 30—after 30 min).

**Figure 2 materials-14-01611-f002:**
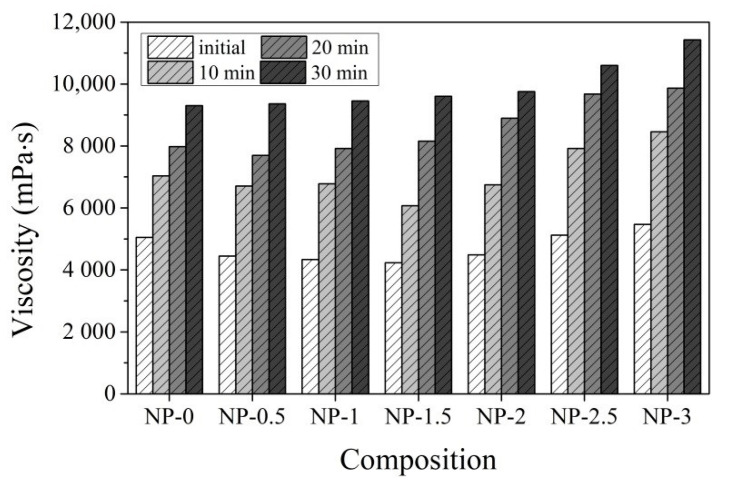
Viscosity of CEM IIN paste (LN) with different dosages of CN (0–3%) depending on time at +20 °C temperature (0—immediately after paste preparation; 10—after 10 min; 20—after 20 min; 30—after 30 min).

**Figure 3 materials-14-01611-f003:**
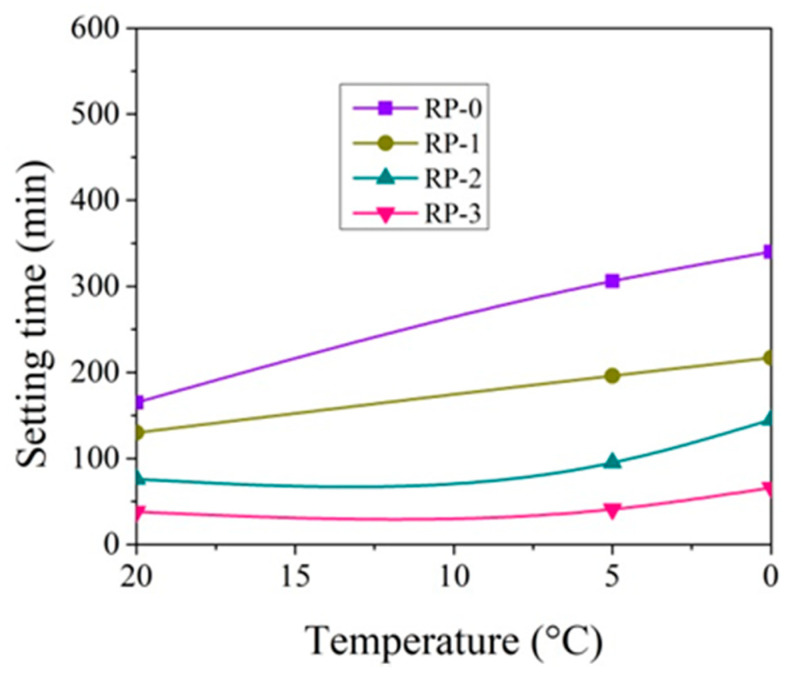
Initial setting times of CEM IIR paste with 0–3% CN at different temperatures.

**Figure 4 materials-14-01611-f004:**
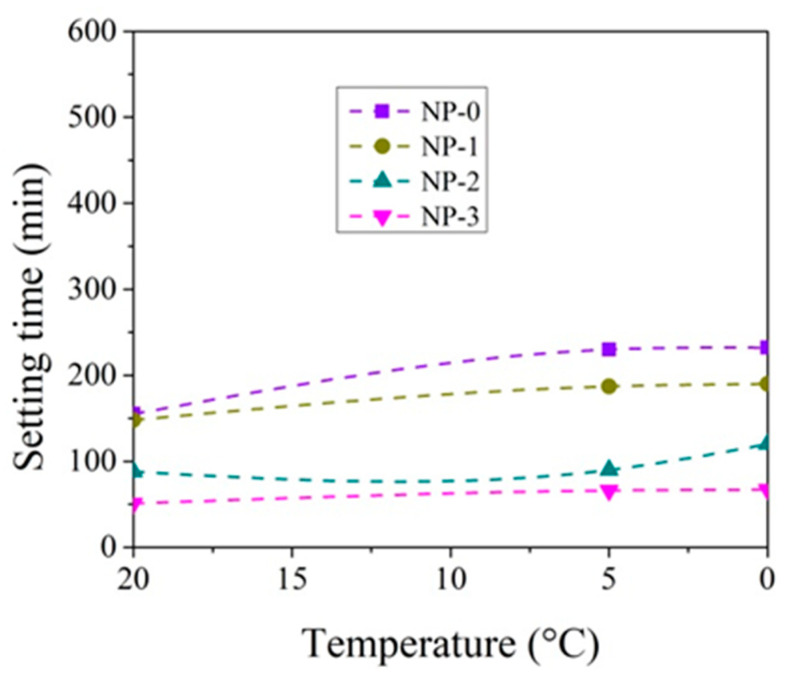
Initial setting times of CEM IIN paste with 0–3% CN at different temperatures.

**Figure 5 materials-14-01611-f005:**
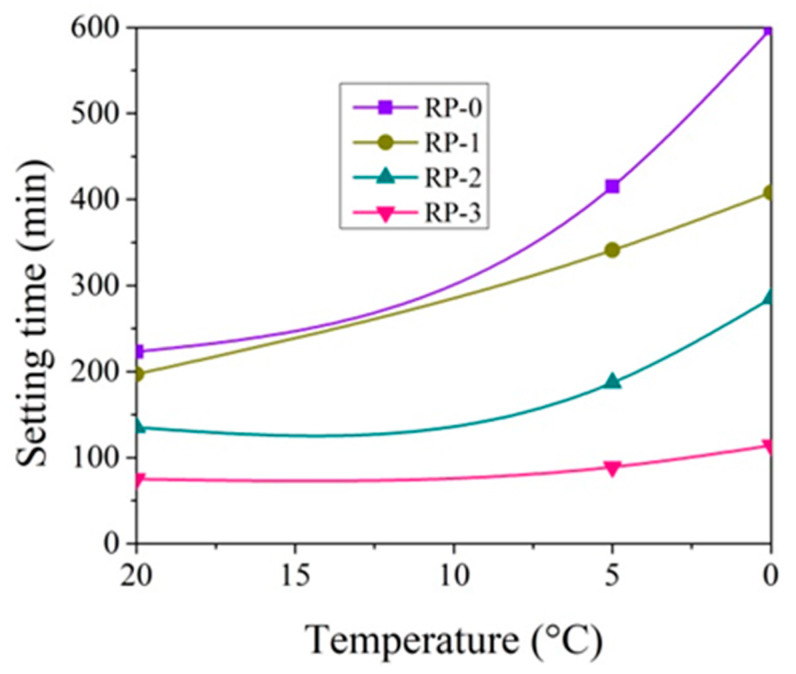
Final setting times of CEM IIR paste with 0–3% CN at different temperatures.

**Figure 6 materials-14-01611-f006:**
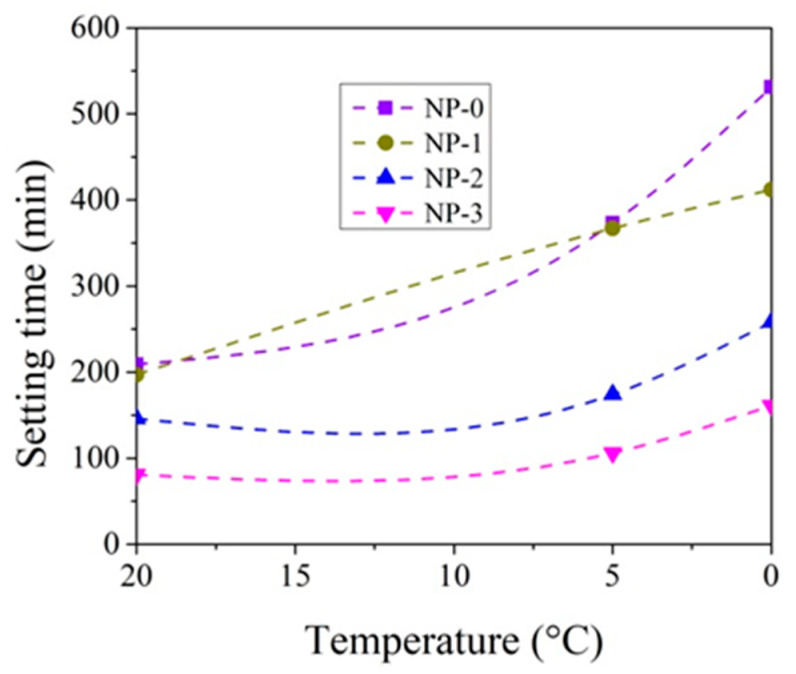
Final setting times of CEMIIN paste 0–3% with CN at different temperatures.

**Figure 7 materials-14-01611-f007:**
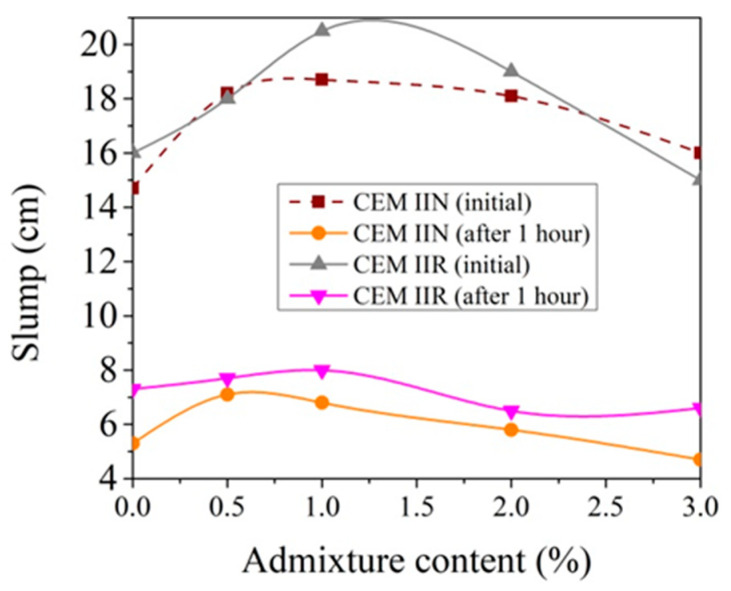
Slump of concrete mixtures with CEM IIR and CEM IIN dependence of CN content (0 h—measured instantly after mixing, 1 h—measured after 1 h).

**Figure 8 materials-14-01611-f008:**
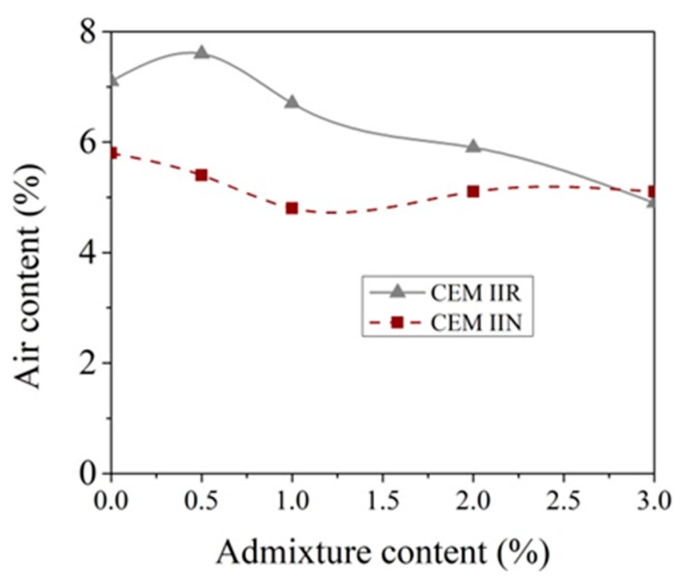
Entrained air content in concrete with CEM IIR and with CEM IIN dependence of CN content (0 h–measured after mixing, 1 h–measured after 1 h).

**Figure 9 materials-14-01611-f009:**
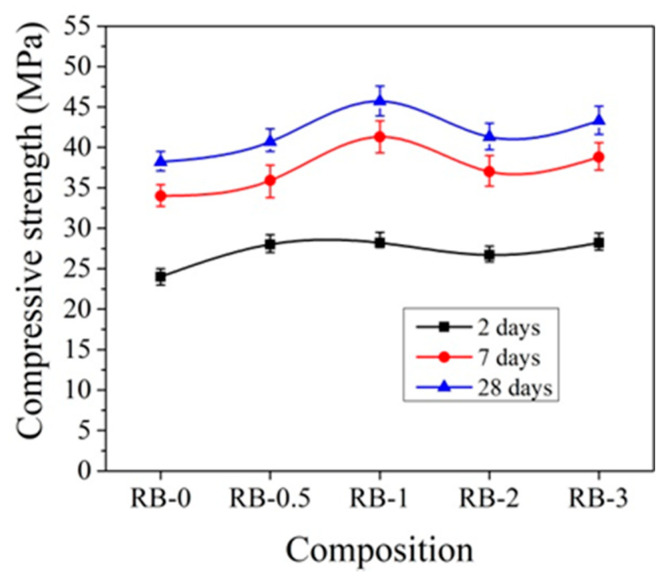
Compressive strength of concrete with CEM IIR samples at +20 °C curing.

**Figure 10 materials-14-01611-f010:**
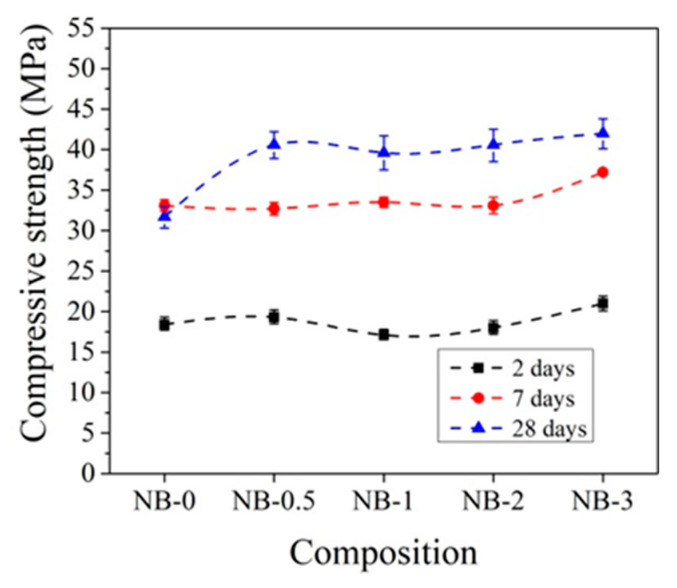
Compressive strength of concrete with CEM IIN samples at +20 °C curing.

**Figure 11 materials-14-01611-f011:**
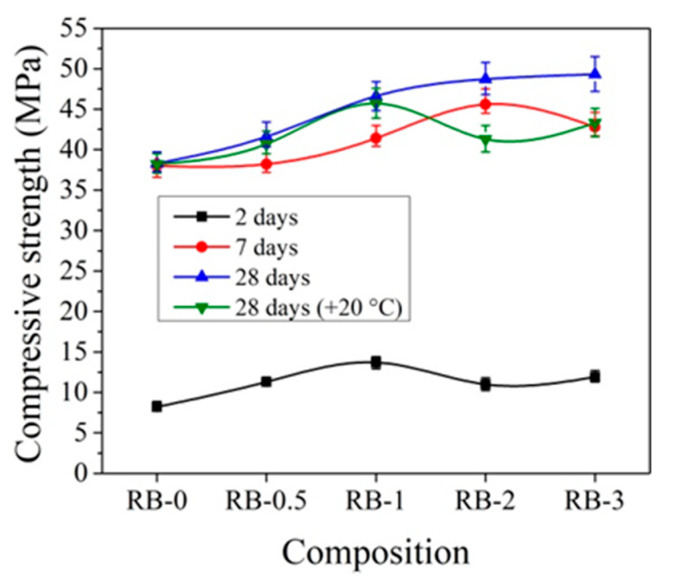
Compressive strength of concrete with CEM IIR samples at +5 °C initial curing (28 days (+20 °C)—reference samples prepared and cured at +20 °C).

**Figure 12 materials-14-01611-f012:**
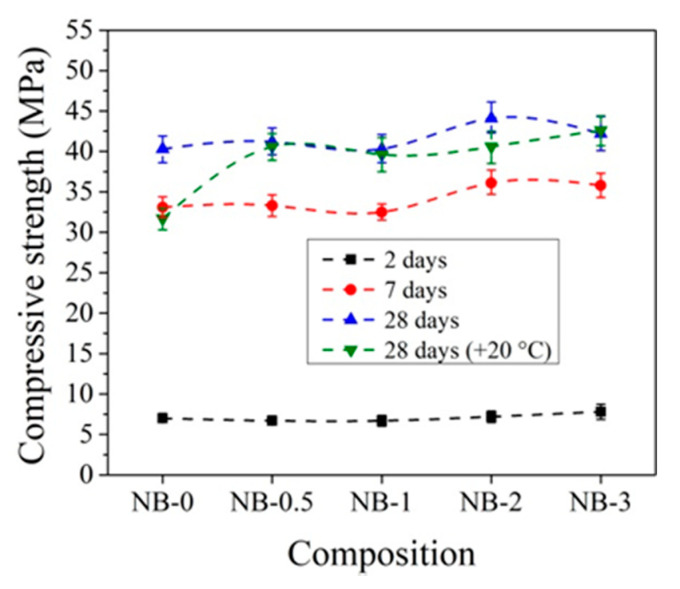
Compressive strength of concrete with CEM IIN samples at +5 °C initial curing (28 days (+20 °C)—reference samples prepared and cured at +20 °C).

**Figure 13 materials-14-01611-f013:**
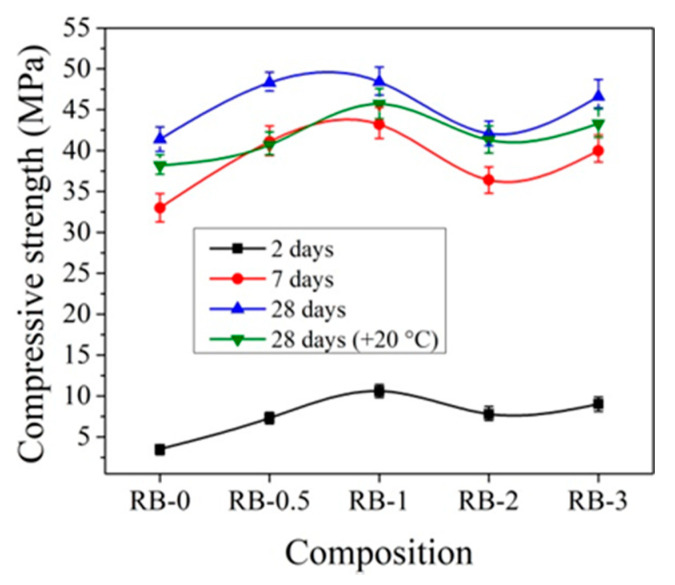
Compressive strength of concrete with CEM IIR samples at 0 °C initial curing (28 days (+20 °C)—reference samples prepared and cured at +20 °C).

**Figure 14 materials-14-01611-f014:**
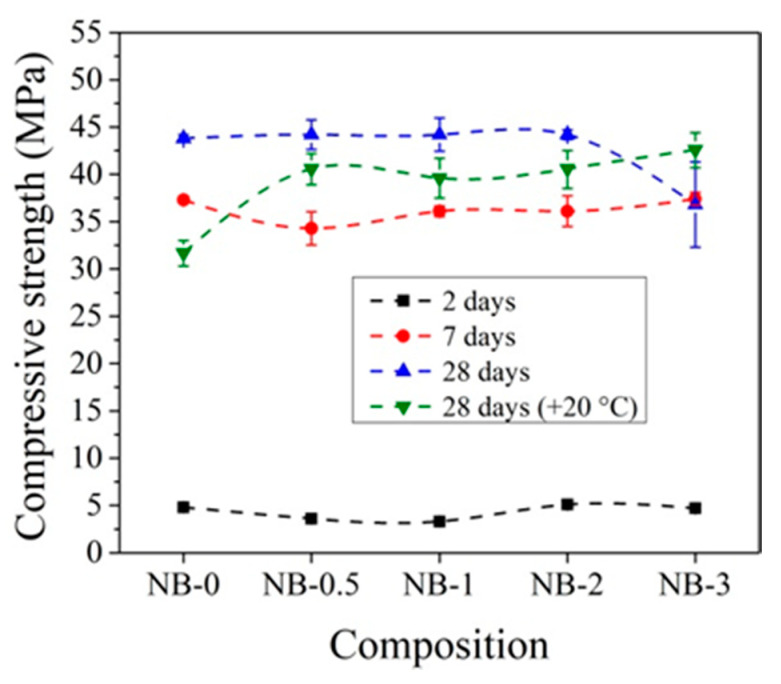
Compressive strength of concrete with CEM IIN samples at 0 °C temperature initial curing (28 days (+20 °C)—reference samples prepared and cured at +20 °C).

**Figure 15 materials-14-01611-f015:**
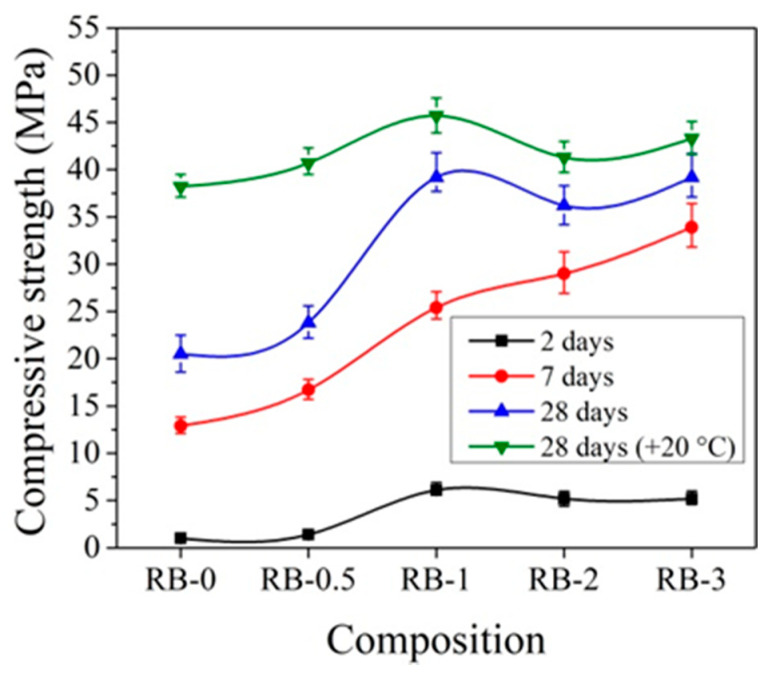
Compressive strength of concrete with CEM IIR samples at −5 °C initial curing (28 days (+20 °C)—reference samples prepared and cured at +20 °C).

**Figure 16 materials-14-01611-f016:**
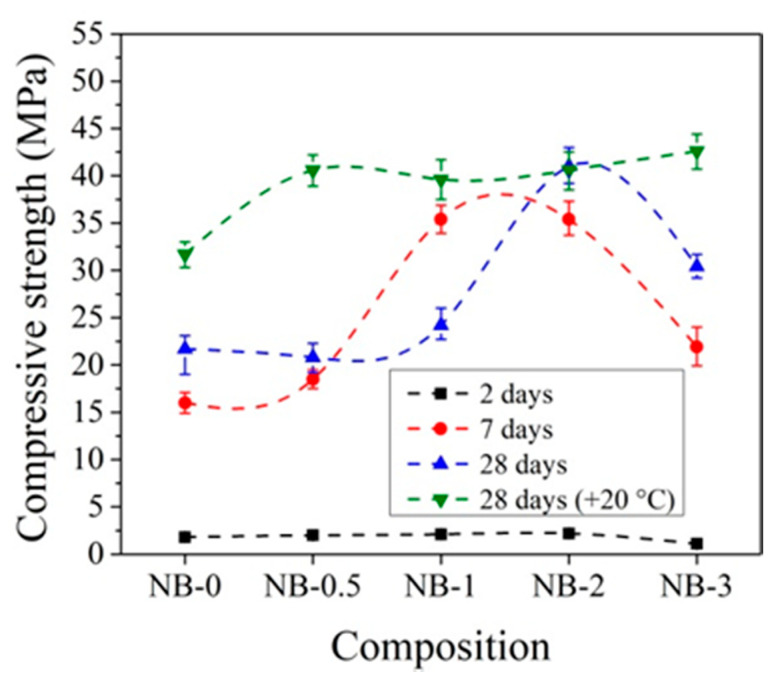
Compressive strength of concrete with CEM IIN samples at −5 °C initial curing (28 days (+20 °C)—reference samples prepared and cured at +20 °C).

**Figure 17 materials-14-01611-f017:**
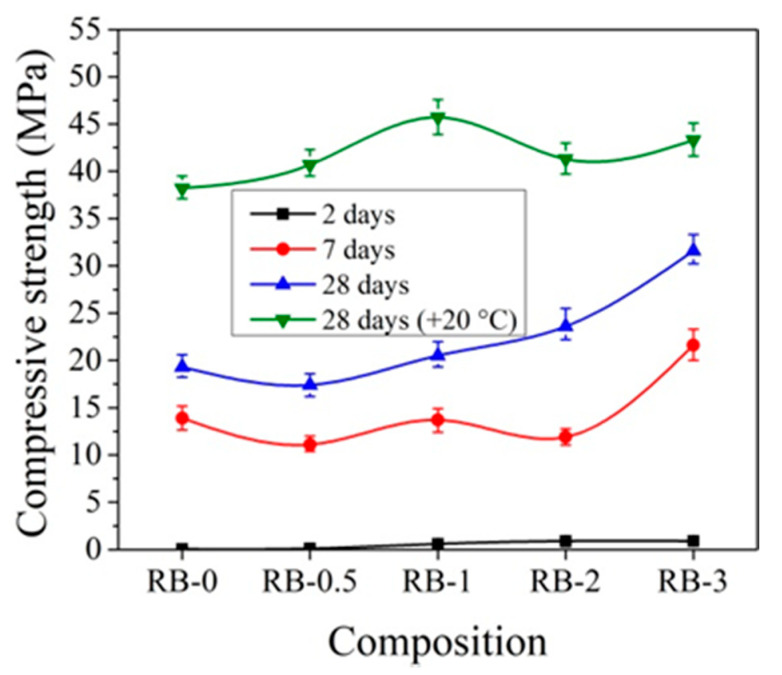
Compressive strength of concrete with CEM IIR samples at −10 °C initial curing (28 days (+20 °C)—reference samples prepared and cured at +20 °C).

**Figure 18 materials-14-01611-f018:**
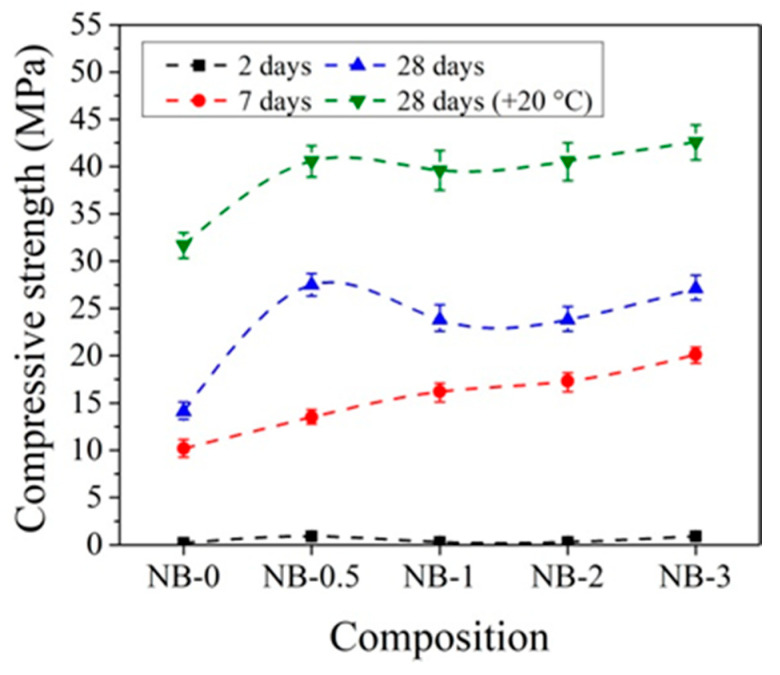
Compressive strength of concrete with CEM IIN samples at −10 °C initial curing (28 days (+20 °C)—reference samples prepared and cured at +20 °C).

**Figure 19 materials-14-01611-f019:**
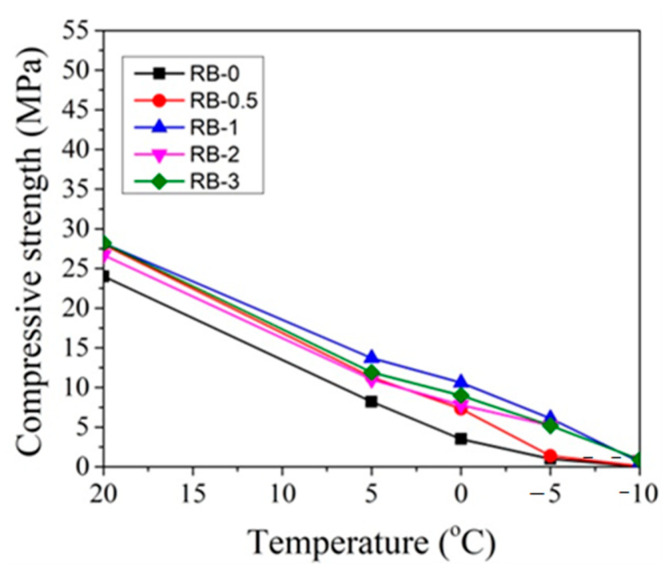
Compressive strength of concrete with CEM IIR after 2 days as a function of initial curing temperatures (°C) for CN dosages in the range from 0% to 3%.

**Figure 20 materials-14-01611-f020:**
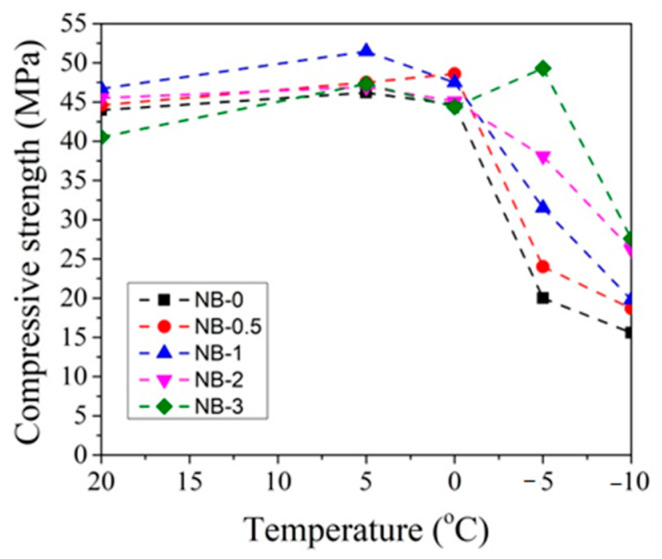
Compressive strength of concrete with CEM IIR after 28 days as a function of initial curing temperatures (°C) for CN dosages in the range from 0% to 3%.

**Figure 21 materials-14-01611-f021:**
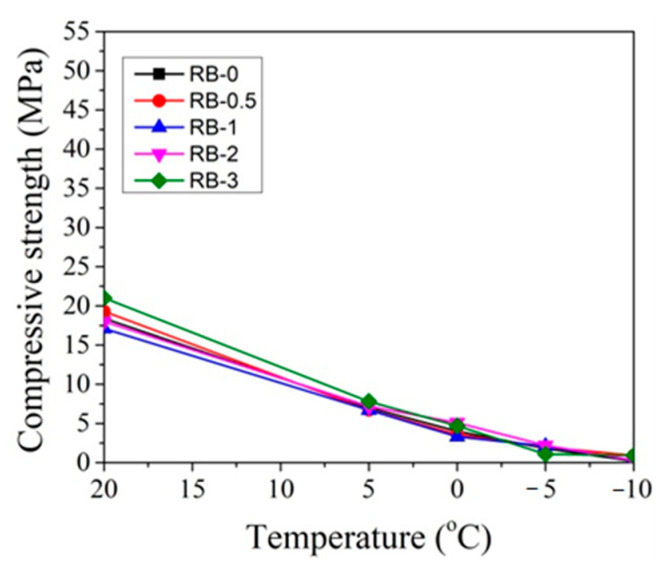
Compressive strength of concrete samples with CEM IIN after 2 days as a function of initial curing temperatures (°C) for CN at dosages from 0% to 3%.

**Figure 22 materials-14-01611-f022:**
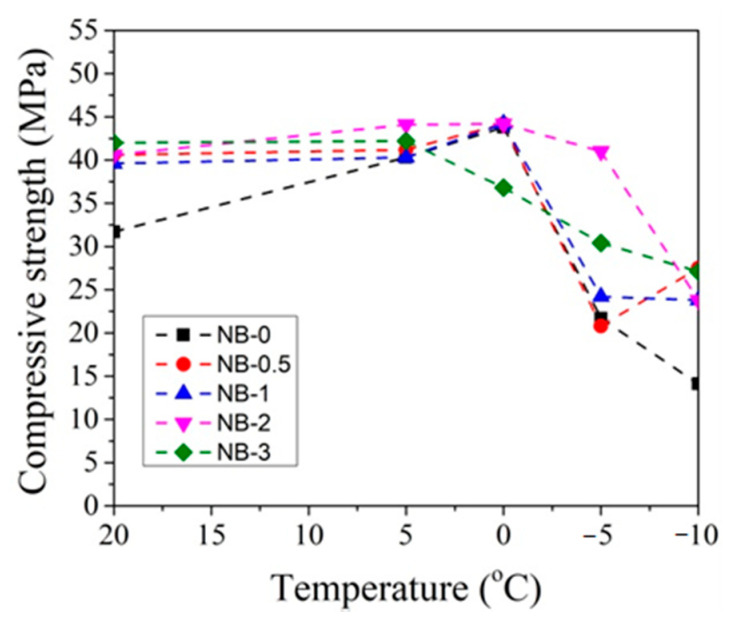
Compressive strength of concrete samples with CEM IIN after 28 days as a function of initial curing temperatures (°C) for CN dosages in the range from 0% to 3%.

**Table 1 materials-14-01611-t001:** Properties of cement.

Cement Type	Marking	Compressive Strength, MPa 7 Days 28 Days		Fineness Blaine, cm^2^·g^−1^
CEM II A-LL 42.5R	CEM IIR	29.9	51.1	4400
CEM II A-LL 42.5N	CEM IIN	23.7	51.2	4080

**Table 2 materials-14-01611-t002:** Mineralogical composition of cement.

Cement Type	Marking	C_3_S, %	C_2_S, %	C_3_A, %	C_4_AF, %
CEM II A-LL 42.5R	CEM IIR	58.7	12.9	6.3	10.79
CEM II A-LL 42.5N	CEM IIN	61.8	8.57	7.1	11.9

**Table 3 materials-14-01611-t003:** Compositions of fresh cement pastes for investigating viscosity and setting times.

Batch	Materials (in Mass%)			
-	CEM IIR	CEM IIN	CN *	W/C
RP-0	100	–	0	0.267
RP-0.5	100	–	0.5	0.267
RP-1	100	–	1	0.267
RP-1.5	100	–	1.5	0.267
RP-2	100	–	2	0.267
RP-2.5	100	–	2.5	0.267
RP-3	100	–	3	0.267
NP-0	–	100	0	0.240
NP-0.5	–	100	0.5	0.240
NP-1	–	100	1	0.240
NP-1.5	–	100	1.5	0.240
NP-2	–	100	2	0.240
NP-2.5	–	100	2.5	0.240
NP-3	–	100	3	0.240

* over 100% of cement.

**Table 4 materials-14-01611-t004:** Quantity (in kg) of raw materials necessary to prepare 1 m^3^ of concrete.

Batch	CEM IIR	CEM IIN	Sand	Gravel	CN	Superplasticizer	Water
RB-0	310	–	925	1005	0	1.55	170
RB-0.5	310	–	925	1005	1.55	1.55	170
RB-1	310	–	925	1005	3.1	1.55	170
RB-2	310	–	925	1005	6.2	1.55	170
RB-3	310	–	925	1005	9.3	1.55	170
NB-0	–	310	925	1005	0	1.55	170
NB-0.5	–	310	925	1005	1.55	1.55	170
NB-1	–	310	925	1005	3.1	1.55	170
NB-2	–	310	925	1005	6.2	1.55	170
NB-3	–	310	925	1005	9.3	1.55	170

**Table 5 materials-14-01611-t005:** Solidification time (min) at temperature −5 °C and −10 °C for CEM IIR paste.

Dosage of CN	Initial Solidification Time at Temperature		Final Solidification Time at Temperature	
	−5 °C	–10 °C	–5 °C	–10 °C
0%	154	137	214	163
1%	138	107	198	150
2%	124	95	190	170
3%	100	100	193	160

**Table 6 materials-14-01611-t006:** Solidification time (min) at temperature −5 °C and −10 °C for CEM IIN paste.

Dosage of CN	Initial Solidification Time at Temperature		Final Solidification Time at Temperature	
	−5 °C	−10 °C	−5 °C	−10 °C
0%	110	122	180	150
1%	132	80	187	123
2%	125	99	180	165
3%	105	100	210	156

## Data Availability

The data are contained within the article.

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
