# Peer review of "Effect of Calcium Nitrate on the Properties of Portland–Limestone Cement-Based Concrete Cured at Low Temperature"

_materials, 2021, doi:10.3390/ma14071611_

Round 1
Reviewer 1 Report
The research is original, novel and important. Text is effective, clear and well organized. The article develops knowledge in the important topic of the influence of additives on mix rheology and concrete properties. The authors analyzed effect of calcium nitrate with dosages from 0 to 3 % of cement mass on the properties of fresh cement paste rheology and hardening processes, as well and strength of hardened concrete at different curing temperatures. The attention to the different behaviour of the mix and the concrete depending on the curing temperature is valuable. The results are analysed in detail with attention to the chemical composition and the chemical reactions that may account for the relationships obtained. This is a very valuable part of the article. The abstract is correct, contains a summary, key findings. Methodology is well described. It is possible to reproduce the tests carried out on the basis of the article. The conclusions are generally correct. Few detailed comments have been presented below.
General remarks:
Air content results - this is unclear, there is no information about the tests or whether the concrete was aerated intentionally (the values obtained indicate this). Do the results refer to the mix or to the hardened concrete? This needs
clarification and perhaps refers to a different value than what the reviewer understands as air content,
The authors in many places use additional (perhaps unnecessary) "-" "()" characters and repeat the "- type" designation, for example: "the same - 0.55." line 142; (-10oC to 20oC) line 147; "density – according to" line 150; CEM IIN – type
line 168, 169 and many other places. This requires consideration of whether it is necessary and perhaps removal. It is the reviewer's opinion that the removal of these marks will improve the readability of the text,
In many other places the authors compare samples with strength, or some other tested quantity. This is understandable in everyday language, but in a scientific article it must be stated very precisely what is being compared with what. The whole section on results and their analysis should be checked in this respect and corrected,
Testing the effect of the additive on small samples is not meaningful compared to a more massive structure where heat transfer to the outside is slower and the concrete heats up to a much higher temperature. It is worth commenting on this topic. Perhaps it is possible to compare the results to those done on larger components?
Detailed remarks:
Abstract: The abbreviation calcium nitrate should not be used in the abstract, but given that it is repeated several times an exception can be made.
The designation CEM II A-LL 42.5 should be completed with the letter R/N.
line 38: "produced from processing raw materials" - doesn't that also require energy and burning fuel. Please explain,
line 53: "showed similar properties" - please specify which properties were tested, including those related to the tightness and durability of the concrete?
lines 60-61: "which freeze water and slows down the hardening processes" - this is somewhat unclear - if the water freezes then hydration stops altogether? It is written that it slows down, this needs clarification,
lines 110,112: "calcium nitrate (CN)" - this is not the first place where the name calcium nitrate appears. The abbreviation should be explained earlier (not just in the abstract) and only once is enough,
tables 1, 2: need to improve layout - widen first column, unify alignment in cells with results, move text 7 days ; 28 days,
line 203: "temperature from 20oC to 0oC (Figures 3 -6)." - Stating the temperature from a higher value to a lower one contradicts the description of the x-axis in the graphs, where from the left is 0 and increases to the right. This reduces the readability of the presentation of the results. It is understandable that values are compared to the more familiar ones obtained at t=20oC and it is difficult or impossible to replace this in the text. It is worth considering the reverse description of the axes on the graphs and inverting the graph,
lines 210,218,219: "to +5oC." "relative to +20oC." - The + sign is unnecessary and makes it difficult to understand why it is given by the authors in one place and not in other places when the temperature is positive? Perhaps it also appears in other places,
figures 3-6: please standardise the scales on the x-axis, the change in diagrams 5 and 6 seems unjustified,
line 271: slump of the concrete mix by 33 mm - Is it worth giving such an accurate value? According to the reviewer, slump cannot be measured with an accuracy of 1 mm. It is worth stating whether the consistency class changes
depending on the admixture dosage,
lines 284-293: The results given raise serious doubts, which are also described in the General remarks. This requires clarification and probably a correction of the text,
figures 9-10: Please uniform the size of the figures and place the description of the Y axis in Fig. 10 on one line,
lines 302,303: "Compared to control samples after 2 days of hardening, the compressive strength of concrete" - Here and in many other places, for example lines: 320,321 the authors compare samples with strength, or some other tested quantity. This is understandable in everyday language, but in a scientific article it must be stated very precisely what is being compared with what,
lines 379-380 - incomprehensible part, needs rewording and clarification,
line 383: "The authors believe that" - this is not a precise statement, it needs to be changed,
line 383: "the reference CN-free reference" - the repetition should be removed, the term CN-free is not in the reviewer's opinion the most accurate but does not need to be changed,
line 406: "No concrete specimens with CEM IIN do not reach" - double negation probably grammatically correct, but perhaps it would be better to write it differently?
figures 19-22 - the temperature is not specified in the axle description,
line 468: "range 20oC – 0oC," - it is not quite clear why the higher value is before? Perhaps it is possible to change this, as indeed 20oC is the reference temperature.
line 474: "temperature interval 20 -0oC" - probably a minor editing error.
Author Response
We deeply appreciate the time and effort you have spent in reviewing our manuscript. Please see the attachment.

Reviewer 2 Report
The article entitled Properties and hydration kinetic of calcium nitrate modified composite Portland cement concrete at low temperature deals with the influence of calcium nitrate on Portland cement pastes at different temperatures. For this reason, I would recommend changing this topic to better reflect, for example, The effect of calcium nitrate on Portland cement pastes at different temperatures. In addition, the term hydration kinetic appears in the current title, but in the article I found no information that I would consider as a true description of hydration kinetic as an equation, etc.
The article is not very scientific but contains a lot of information for practical use for other research, which I find beneficial. I would recommend, for example, to write it at the end of the introduction that the article is focused on practical information.
The results are described on pages 4 - 18, but in such an extensive set of information. There are only 5 references to the literature, which certainly does not correspond to current trends. In fact, this article does not contain any discussion, but only a very comprehensive description of the data, where each measured value is evaluated several times and presented in different ways. The whole section of results should be changed and supplemented by a real discussion containing a lot of references to the literature. The data processed in this way are interesting, but in the long description it is very difficult to find your way around and it is difficult to find the right and complete conclusion. The explanation of the reasons why the effects occur is usually missing or hidden in the long text or is insufficiently explained. E.g. page 5 states: “We can conclude that lower amounts up to 2% of CN can work as plasticizing admixture but higher amounts more than 2% demonstrate the effect of increasing viscosity. This may be due to the increase in paste’s temperature - with increase of CN amount in the paste, the temperature of paste increases significantly by high CN dosage. This effect has been described by Kicaite et al. [38] ". This is one of the few explanations in the article, but in general the viscosity decreases with temperature. This leads to a very special phenomenon which should be explained in the article and not just by quoting another work.
Page 3 states that more aggregate species have been used, but the amount used is not indicated. The results of viscosity and innitial setting times indicate that they were measured on pastes. However, further results are given as measured for concrete mixtures. According to the measured results and the information provided, I would guess that all the results were measured for pastes, but it should be clearly stated and possibly explained why something was measured this way and something different.
Page 3 also states "The w/c ratio in cement pastes was 0.24 for CEM IIN and 0.267 for CEM IIR pastes." However, I do not know why each paste had a different w/c. The conclusion states "The CEM IIR-type cement granulometry and higher w/c required to achieve normal consistency caused by smaller cement particles of CEM IIR compared to CEM IIN, can induce a more active reaction of CN with the cement minerals C3A and C3S. The reason for the different amount of water used could be to try to maintain the same consistency, but this must be explained on page 3.
Section 4 listed as a discussion contains three lines without any discussion. I propose to call section 3 as results and discussion, and the real discussion must be supplemented.
The article is missing the founding section. This section is not mandatory, but it is usually the most important section for the author's employer. I recommend the author to check it.
Author Response

(The authors gave the same response as above.)

Reviewer 3 Report
The reviewed manuscript describes the possible application of Portland-limestone cement with the calcium nitrate addition for cementing at low temperatures. It is an interesting problem that can find practical use in the industry. Certain issues in this paper, however, require some clarification. The manuscript contains various errors and incorrect wording that hinder a reader from a proper understanding of the text. The most important problems are the test methods inaccurate descriptions and the results discussion section, in the which the information present in the literature was only briefly referred to.
Below is a detailed list of comments:
1. The title is incorrect. I propose to replace it with: “Effect of lithium nitrate on the properties of portland-limestone cement based concrete cured at low temperature.
No studies on hydration kinetics are included in this manuscript.
2. The wrong symbol of degrees denotation was used in the whole paper, it should be: "°" e.g. L17.
3. L18 "(CEM II A-LL 42.5 and 42.5 N)" - missing "R".
4. L33. "cement hydration products matrix" - enough "cement matrix".
5. L57 for the sake of clarity, I would not use the term "retention" in this form (when describing the effect of limestone, "retention" is usually referred in relation to "water retention" [16]).
Moreover, there are opposing theories that could be mentioned, e.g.:
Jambor, J. Influence of 3CaO.Al2O3·CaCO3.nH2O on the structure of cement paste. Proceedings of the 7th ICCC, Paris, France, 30 June-4 July 1980, 487-492.
6. L59-62: sudden mental leap. Sentence of concreting at low temperature is unrelated to the previous text. This should be elaborated on (e.g. refer to the heat of hydration).
7. Section 2. Materials and Methods. The description is incorrect. Tested paste temperatures issue were not mentioned at all, although they are important for further study. It is not known how the planned temperature of the pastes was ensured during the tests - especially the temperature below 0 °C.
The standard testing of the setting time (EN 196-3 - the Authors do not refer to any standard) with the Vicat device assumes that the samples are placed in a water bath to ensure appropriate conditions - thus, it cannot be carried out without modification at a temperature below 0 °C.
The conditions in the freezing chamber for storing concrete samples should also be presented in more detail - what was the humidity in it? After what time were the samples demoulded? Due to the possibility of water freezing - was there any dimension change?
8. Section 2. Were both cements from the same source and were they produced in the same period? So, can it be presumed that both are made of the same clinker? Such information would be useful.
9. Table 1: describe what "> 90 μm" means - give the appropriate column title.
10. L122-124. What kind of coarse aggregate was used?
11. L145-150. It is essential to determine how many times the individual measurements were repeated for each paste/concrete test (especially in the consistency and compressive strength test)?
12. I would suggest introducing tables with the individual sample composition, which should be properly marked. Currently, the markings on the graphs (legends) in Section 3 are not easy to understand, because they are not explained anywhere (samples "0" or "N (0h)"), or the units are missing (the reader can only guess their meanings) - the legends in Figures 1-8 need to be corrected.
13. The figures should be standardized - the colouring of Fig. 19-22 is clearly different from the previous ones.
14. Figure 1 and 2: Describe the X axis.
15. L152. Rename the subsection title: Viscosity of the cement paste with CN.
16. L171-172. Reference should be made to the relevant literature to explain the role of C3A and C3S.
17. L180-182 Correct all "-" to "–" (this error is repeated).
18. L193-194: „… up to 2% of CN can work as plasticizing” – too hasty conclusion. The studies on only one rheological parameter - viscosity are presented here. I would suggest that yield stress be tested further in the research. To formulate this conclusion, it would also be useful to present results of previous adequate consistency tests (eg. according to EN 196-3). Remember that the changes caused by the plasticizer should be stable over a certain period of time.
19. Figure 3-6: Set the same scale range of X and Y axis for all graphs. Alternatively, I would suggest presenting the results of the initial and final setting time tests together on the column graph, making the comparison easier.
20. L242-262 as the authors themselves note, below 0 °C it is difficult to define the basic mechanism of "binding". In the case of the reference samples, the freezing of water will be more likely. Therefore, it should not be referred to as binding. To distinguish it from the cement binding (hydration), for example, I would suggest using the term of "paste solidification process" or "setting time test results", but definitely not "setting time". The conclusion from this, however, is that the used method does not allow proper test of cement setting (in terms of hydration) at a temperature below 0 °C (Comment 7). The presented results of the concrete compressive strength test show that the cement does not bind but only freezes.
The authors should write something more about the problems related to water freezing during binder hydration in the introduction. It is important due to the interpretation of the test results.
21. Figs 7-18. The results scatter should be presented using the error bars on the graphs – this is necessary for the correct interpretation of the results.
22. L264-276: Please check and correct if necessary. The period of time after which the described tests were performed is unknown.
What is the difference in the slump for CEM IIN with 0.5% and 1% CN after 0h - 7 mm? - there is hardly any difference in the chart.
23. Figs. 11-18. why the “28dp-reference samples” results are different in all charts. They should be constant for one type of cement. They also do not match to the 28-day results in Figure 9-10 - from the description it does follow that they are the same samples.
24. Section 4: the results should be discussed in relation to previous studies; the authors do not cite any (even their own [38]) in this section. Many of the excerpts from the section 3 are de facto also a discussion of the results - including description of the C3A and C3S effect (comment 16).
I believe that in sections 3 and 4 the authors should relate the obtained results to the literature data to a greater extent and rethink the structure of these sections.
25. L486-487: “can induce a more active reaction of CN with the cement minerals C3A and C3S, explaining the more rapid increase of viscosity.” – in this work it is an unconfirmed hypothesis. There were no studies on the effects of C3A and C3S – therefore this fragment should be deleted.
26. L488: „plasticizing effect in the cement pastes “ - on the basis of the same viscosity test, I do not think that we can talk about the plasticizing effect in the cement pastes case (comment 18).
28. L489: „The plasticizing effect does not change during the first hour.” - this statement may be misleading. Regarding the previous sentence „…plasticizing effect in the cement pastes…” – the pastes viscosity after one hour was not tested. The concrete slump itself decreases within one hour.
28. L496-498: “At temperatures -5oC and -10oC 3% CN can shorten the initial setting time 1.54 times and 496 1.37 times, respectively, in case of CEM IIR-type paste, while analogous numbers for CEM 497 IIN-type paste are 1.05 and 1.22 times.” - this is incorrect. Setting time was not investigated here (see comment 20).
29. Author Contributions should be supplemented. Who was responsible for the visualization (i.e. figure preparation)?
30. Provide the required information about the source of funding.
Author Response

(The authors gave the same response as above.)

Round 2
Reviewer 2 Report
The discussion has been added and the article now better meets the standards for publication in the journal. I still don't know from the article if aggregate was used. If so there is no information on its quantity. If the mixture has been mixed according to a standard, this standard must be stated. This must be clear otherwise the results are worthless. Other comments were taken into account.
Author Response
Please see the attachment.
Best regards
Asta KiÄŤaitÄ—

Reviewer 3 Report
Some significant and beneficial changes have been made in the manuscript, but the Authors have not responded to all the Reviewer’s comments (notes 5 and 12 from the list below). Also, not all changes made in the manuscript have been marked (e.g. additional sentence in L163-164). The interpretation of the setting time research results for pastes at -5 °C and -10 °C raises serious doubts. No evidence has been provided to prove that the setting time was actually measured under the conditions by the indicated method (Note 7).
Below is a detailed list of comments:
- Regarding comment 2: The wrong symbol of degrees denotation was used in the whole paper, it should be: "°" e.g. L17.
As the reviewer suggested, the symbol of degree was corrected in the whole paper
Such a correction is not fully correct. The correct degree symbol was used e.g. in L60, 125; incorrect, e.g. in L61, 78, 80, 82. The space in the temperature notation, e.g. L127; instead of "+ 20°C" it should be "+20 °C". Please check the entire manuscript and correct it.
- L131-137: please specify at which needle penetration the initial and final setting times were determined - have the requirements such as in EN 196-3 been adopted in this respect?
- Regarding to comment 10. L122-124. What kind of coarse aggregate was used?
Pubble 20 mm for investigations was used.
What is "pubble"? Is it gravel aggregate? It is necessary to write about the aggregate type used in the research (L122-124).
- Regarding to comment 13. The figures should be standardized - the colouring of Fig. 19-22 is clearly different from the previous ones.
Figures are corrected.
Figures were not corrected.
- Regarding to comment 17. L180-182 Correct all "-" to "–" (this error is repeated).
No response from the Authors. There are numerous errors of this kind in the paper.
- Regarding to comment 19. Figure 3-6: Set the same scale range of X and Y axis for all graphs. Alternatively, I would suggest presenting the results of the initial and final setting time tests together on the column graph, making the comparison easier.
Figures are redrawn again.
Contrary to the recommendation, the scale of the Y axis has not been standardized.
- Regarding to comment 20. L242-262 as the authors themselves note, below 0 °C it is difficult to define the basic “setting” mechanism. In the case of the reference samples, the freezing of water will be more likely. Therefore, it should not be referred to as setting. To distinguish it from the cement hardening (hydration), for example, I would suggest using the term of "paste solidification process" or "setting time test results", but definitely not "setting time". The conclusion from this, however, is that the used method does not allow proper test of cement setting (in terms of hydration) at a temperature below 0 °C (Comment 7). The presented results of the concrete compressive strength test show that the cement does not bind but only freezes.
The authors should write something more about the problems related to water freezing during binder hydration in the introduction. It is important due to the interpretation of the test results.
Thanks for the comments. The text of the article is corrected.
When both of the cement pastes’ setting times were tested at -5oC, significantly reduction of setting times compared to data at 0oC was obtained (Tables 2 and 3). When the ambient temperature is below 0oC, water freezing processes begin to predominate and cement hydration processes significantly slow down, making it difficult to accurately separate the binding processes according to commonly used test procedures. In this case, different effects are seen for different types of cements. A shortening of the initial setting time is observed at -5oC for CEM II R paste. As C2S amount is by 1.5 times higher in this cement composition than in CEM IIN, the CN efficiency as a set accelerator in paste with this cement is more pronounced [25]. Meanwhile, we do not see such effect for CEM IIN paste. Compared to an investigation at 0oC, just 1% of CN has an effect on the initial setting time of CEM IIN paste2 and 3% of CN did not reduce the freezing point of the water [45]. In the case of the final setting, we also observe that the use of CN for CEM IIR paste shortens the final setting time.
I still find the author's interpretations as questionable due to the misleading vocabulary used.
The authors fail to provide any evidence that the cement sets was measured by Vicat method under the described conditions (-5 and -10 ºC - according to Tab. "2" and "3"). Therefore, writing about the "setting time" or "initial setting time is misleading. As mentioned before, the gradual freezing of the mixing water may be responsible for the needle penetration reduction, not the cement setting - i.e. the freezing speed is measured. For this reason, the results presented in Tables 3 and 4 may be "false" (and so they can be called )
It would be useful to perform other tests confirming both the possibility of cement hydration and the efficiency of this process at temperatures under 0 °C. For example, the hydrating cement reaction degree under such conditions or hydration kinetics (e.g. by an isothermal method) could be examined. It is advisable to compare the effect of setting and freezing of the cement paste on its consistency. This could be an interesting direction for further research.
- Table 2 and 3 - wrong numbering. There are two tables 2 in the paper - pages 3 and 8.
- Regarding comment 21. Figs 7-18. The results scatter should be presented using the error bars on the graphs – this is necessary for the correct interpretation of the results.
In studies, an arithmetic mean was calculated.
The comment remains the same: please present the results scatter.
- Regarding comment 22. L264-276: Please check and correct if necessary. The period of time after which the described tests were performed is unknown.
What is the difference in the slump for CEM IIN with 0.5% and 1% CN after 0h - 7 mm? - there is hardly any difference in the chart.
We present the results of the study. It can be seen that only a small difference.
It is still not known exactly which slump studies are described - it is necessary to refer to the time of the tests (where the Authors describe the tests after 0h and where after 1h?).
- Regarding to comment 26. L488: „plasticizing effect in the cement pastes “- on the basis of the same viscosity test, I do not think that we can talk about the plasticizing effect in the cement pastes case (comment 18).
CN content of 0.5% and 1% increases the slump of concrete. The slump of concrete mix regardless of the type of cement, decrease when CN amount increase above 1.5%. These slump results for concrete correlates well with viscosity studies on pastes.
I understand the Author's opinion, but I would still recommend changing this statement. I believe that in the manuscript the correlation between the paste viscosity and the concrete slump has not been discussed in detail enough to clearly lead to this statement (only one sentence L310-311 was devoted to this). For this reason, in the discussion, I would recommend more precisely comparing the results of the viscosity and slump test or changing the wording of this conclusion.
It can be started:
CN content of 0.5% and 1% by influencing on the paste viscosity may cause the plasticizing effect of concrete mix, which is manifested by increased slump.
- Regarding to comment 28. L489: „The plasticizing effect does not change during the first hour.” - this statement may be misleading. Regarding the previous sentence „…plasticizing effect in the cement pastes…” – the pastes viscosity after one hour was not tested. The concrete slump itself decreases within one hour.
„The plasticizing effect does not change during the first hour.” - this statement may be misleading. Regarding the previous sentence „…plasticizing effect in the cement pastes…” – the pastes viscosity after one hour was not tested. The concrete slump itself decreases within one hour.
No response from the Authors - only the reviewer's comment was quoted
- Regarding to comment 28. L496-498: “At temperatures -5oC and -10oC 3% CN can shorten the initial setting time 1.54 times and 496 1.37 times, respectively, in case of CEM IIR-type paste, while analogous numbers for CEM 497 IIN-type paste are 1.05 and 1.22 times.” - this is incorrect. Setting time was not investigated here (see comment 20).
The setting time of CEM IIR and CEM IIN pastes at temperatures -5oC and -10oC was investigated and presented in 2-3 table. Describing the setting time additional discussion is provided, with the justification that hydration of cement takes place even at low temperatures, although its pace is very slow. Therefore, we believe that the research data provided by us should be reflected in the conclusions
In view of the doubts presented in comment 7, I consider this conclusion wording as incorrect. It may mislead the reader. Due to the possibility of the sample’s freezing, the results of the setting times tests in Table 2-3 may be "false". It is not known what role in obtaining them can be attributed to freezing and what to a real setting. Resolution of this could be an interesting research topic. Currently, however, I believe that the authors should not write about the "setting time" regarding the tests at -5 °C and -10 °C. When writing about the results of these tests (also in the conclusions), it should be mentioned that the freezing phenomenon could have had a significant role in obtaining them.
Author Response
Thank you for review.
Please see the attachment.
Best regards
Asta KiÄŤaitÄ—

Round 3
Reviewer 3 Report
The reviewed manuscript has been significantly improved. However some minor errors still occur in it.
1.There are some editorial errors in the text:
- other formatting for some paragraphs, e.g. larger line spacing in L62-73, 123-125, 228-236, 298-307.
2. L149 „intrusion” change to „penetration”
(For the future, I recommend applying the EN 196-3 requirements for the setting time testing by the Vicat apparatus: initial setting time - distance between the needle and the base-plate = 6 ± 3 mm; final stetting - time needle penetrates < 0,5 mm)
3.L144.the comment to table 3 should be placed directly below it in L143 - the same applies to table 4.
4. Author Contributions: regarding to the authors previous replies (to comment 29 from the first review) - please add the information: "Visualization, A.K.".
L650 - lack of dot after "I.P" initials.
5.L656-657 Provide information about funding source according to requirements of the Instructions for Authors - post them as "Funding" or "Acknowledgments"
Author Response
We appreciate the Reviewers' warm work earnestly.
Please see the attachment.
Best regards
Asta KiÄŤaitÄ—
